# A new stress field intensity model and its application in component high cycle fatigue research

SongSong Sun [ID]*

College of Automobile and Traffic Engineering, Nanjing Forestry University, Nanjing, Jiangsu, China

* sunsong1987@126.com

**Data Availability Statement:** All relevant data are within the paper.

**Funding:** This study is finally supported by The National Natural Science Foundation of The Jiangsu Higher Education Institutions of China under the Grant Number of No. 18KJB460019.

## Abstract

Fatigue limit load is one of the most important and concerned factors in designing and manufacturing critical mechanical parts such as the crankshafts. Usually, this governing parameter is obtained by experiment, which is expensive, time-consuming and only feasible in analyzing the case of simple structure. Still, there's a big obstacle to clear to get the fatigue limit load of a sophisticated structure effectively and efficiently. This paper applied the stress field intensity theory to make quick component fatigue limit load predictions. First, the field diameter of a given crankshaft was determined based on its limit stress state and a stress distribution fitting approach. Then, this parameter was used to predict the high-cycle bending fatigue limit load of a new crankshaft composed of the same material. Finally, a corresponding experimental verification was conducted to evaluate the accuracy of the predictions. The results indicated that the original stress field intensity model may not be suitable due to the errors in the predictions, which can be attributed to the structural features. The new model proposed in this paper can provide higher accuracy in quick fatigue load prediction, making it superior to the traditional model in engineering application.

## 1 Introduction

Crankshafts, one of the most important components employed in modern internal combustion engines, are subjected to various dynamic loads during operation. According to the fatigue damage theory [1–3], the damage induced by these loads will accumulate until the crankshaft eventually fractures. Accordingly, the serve life of a crankshaft is limited to a certain number of cycles [4]. Thus, compared with the common fatigue property evaluation parameter (usually the fatigue life under a given load) [5–7], it is more important to correctly evaluate the high-cycle fatigue limit load of a crankshaft under a specified fatigue life [8–9](which is always determined to be $10^7$ cycles).

For most engine parts, the high-cycle fatigue strength is obtained through standard fatigue testing, which is feasible to conduct on small engine parts, such as connecting rods or camshafts. However, for structures with complicated shapes, such as crankshafts, standard fatigue testing is time consuming and expensive. In addition, other factors such as the manufacturing process [10–12], control strategy [13–15], surface strengthening [16–17] will also create

**Competing interests:** The authors have declared that no competing interests exist.

substantial impacts on the fatigue strength of a part, which will result in some errors during the fatigue strength prediction.

In recent years, creative studies have been performed in the field of fatigue property prediction. Among which Marsavina researched the notch fatigue property of AM50 magnesium alloys and found that different type of notch will affect the fatigue strength in different degrees [18]. Linul researched the low cycle fatigue property of aluminium alloy foams and discovered that the presence of non-homogeneities in the cellular structure had a significant influence on the fatigue life of the foam [19]. Berto assessed the accuracy and reliability of the Theory of Critical distances (TCD) and the strain energy density (SED) approach in estimating the lifetime of plain and notched specimens, the results obtained demonstrate that both the TCD and the SED approach can provide highly accurate fatigue life estimation [20]. Sadowski studied the fatigue response of the hybrid joints and found that the cyclic loading will result in plastic damage in the aluminium [21].

Among the reported methods, the stress field intensity approach proposed by Yao-Weixing is considered to be one of the most effective methods for fatigue failure research [22–23]. This approach has a similar expression as the theory of critical distance (TCD) approach [24]. However, the TCD considers that the stress at any point within the stress concentration area has the same impact on the whole process of fatigue, whereas the stress field intensity approach considers that different points have different contributions. According to the definition of the stress field intensity approach, for a given notched component, if the stress field intensity within the vicinity of the notch root is equal to the stress of a smooth specimen, the fatigue lives of the two specimens will be the same. The expression of stress field intensity is as follows:

$$\sigma_{FI} = \frac{1}{V} \int_{\Omega} f(\sigma_{ijk}) \varphi(r) dV \tag{1.1}$$

where $\sigma_{FI}$ is the stress field intensity value, $f(\sigma_{ijk})$ is the stress damage function (usually Von Mises stress), $V$ is the damage volume, $\Omega$ is the stress field range and $\varphi(r)$ is the weight function, which shows the influence of every point within the stress field. The main features of this weight function are as follows:

1. When $0 \leq \varphi(r) \leq 1$, $\varphi(r)$ is a monotonically decreasing function of $|r|$, in which $r$ is the distance between a particular point and the maximum stress point;

2. When $r = 0$, $\varphi(r) = 1$, because the maximum stress point has the greatest impact on the fatigue process;

3. When the stress gradient in the component is 0, $\varphi(r) = 1$. Because the smooth specimen has no stress concentration, any point has the same impact on the fatigue process.

In the application of this approach, the weight function has many expressions. In general, the weight function is considered to be related not only to the distance, but also to the stress gradient. The most popular expression of this function is as follows:

$$\varphi(r) = 1 - |\chi| r \tag{1.2}$$

Where $\chi$ is the stress gradient of a point within the stress field. The definition of this stress gradient is as follows:

$$|\chi| = \frac{1}{\sigma_{max}} \frac{d\sigma(x)}{dx} = \frac{\sigma_{max} - \sigma_r}{\sigma_{max} r} \tag{1.3}$$

Similar to the definition of the TCD, the range of $r$ in the application of the stress field intensity approach is called the field diameter. The definition of this parameter is as follows:

$$\sigma_{FI} = \sigma_b \tag{1.4}$$

Where $\sigma_{FI}$ is the stress field intensity of a given component in its limit condition and $\sigma_b$ is the fatigue limit of the material, which can be determined by experiments with standard smooth specimens. Thus, for a given component under its limit load, when the value of the stress field intensity of the component is equal to the fatigue limit of a smooth specimen, the corresponding integration range of $r$ is the field diameter.

Using this stress field intensity approach, Yao- Weixing predicted the fatigue lives of several components and compared the fatigue limits under different types of loads. Liu Gang predicted the fatigue lives of some welded joints and evaluated the impacts of their structural parameters [25]. Zhou-Shangmeng obtained the diameter of the stress field by experimenting on specimens with different circular holes and predicted the fatigue stress of a steel bridge [26]. Sha-Yunfei combined the stress field intensity approach with the traditional local stress-strain approach and predicted the fatigue lives of some components, a comparison between their predictions and experimental data showed that this approach may achieve good accuracy [27]. Ramezani compared different weight definitions and proposed an advanced volumetric method that can increase the accuracy of predicting the fatigue lives of smooth specimens [28]. Zhao Dan applied the stress field intensity approach to predict the fatigue lives of wire ropes, wherein the theoretical values and the test results were close to each other [29].

In previous research, the stress field intensity approach has usually been applied to evaluate components with simple structures. For a complicated engineering part, such as a crankshaft, the stress distribution cannot be directly determined through theoretical analysis, which creates some difficulties in actual application. Besides, the stress state of this solid part is usually multiaxial even under the uniaxial load. While the stress field intensity approach is just considered to be one of the most useful approaches in researching this problem [30].

In this paper, a crankshaft fatigue limit load prediction model which is based on the stress field intensity approach is proposed, and corresponding experimental verifications are conducted. The major novelty in this manuscript is the modified model based on the stress gradient. Compared with Yao's stress field intensity model, this modified model can predict the fatigue limit load more accurately, which make it more suitable for actual application.

## 2 Method

### 2.1 Stress distribution approach

According to the definition of the stress field intensity approach, the fatigue limit load of any component can be easily computed if the following parameters are known: the material fatigue limit, the stress distribution function and the field diameter. Among these three parameters, the fatigue limit of the material can be obtained through an experiment with a standard smooth specimen. The field diameter is always considered to be a material property constant, which means that the components made from the same material will have the same field diameter. It's feasible to determine the stress distribution function through direct theoretical analysis. However, for a part with a complicate shape such as a crankshaft, it's difficulty to determine the distribution function easily. In a previous study, we applied the finite element method in such occasion, which can determine the stress at the element node conveniently. However, for the point outsides the element nodes, the value can't be determined directly.

In order to solve this problem, Andrea Spaggiari proposed a stress gradient based approach to fit the stress distribution function of the 2D infinite notched plate [31]. In this paper, the object of research is a crankshaft, which has the finite amount of volume. Thus we proposed a combined finite element analysis and inverse function approach to fit the stress distribution function, as follows:

Step 1: Perform a finite element analysis of the crankshaft and record the values of the stress at the nodes along the damage path(from the fillet of the crankpin to the main bearing).

Step 2: Assume the stress distribution function is an inverse function with the following expression:

$$\sigma(r) = \frac{Ar + B}{r + c} \tag{2.1}$$

where $\sigma(r)$ is the stress distribution function, $r$ is the distance from the maximum stress point, and A, B and C are constants determined by the stress distribution.

Step 3: Taking the values of A, B and C as variable parameters to fit this distribution function with the stress of the nodes obtained in Step 1, the main constraint conditions of the fitting goal are as follows:

1. The values of the maximum stress obtained based on the finite element method and the distribution function are the same:

$$\sigma(r = 0) = \sigma_{max} \tag{2.2}$$

2. The sum of the relative difference between the stress value obtained by the finite element approach and the distribution function is the minimized. The definition of the relative difference is expressed as follows:

$$f = \sum_{i=1}^{n} \left| \frac{\sigma_{FE} - \sigma_i}{\sigma_{FE}} \right| \tag{2.3}$$

where f is the relative difference percentage, and $\sigma_{FE}$ and $\sigma_i$ are the stress values obtained by the finite element approach and the distribution function respectively.

## 2.2 Prediction process

The field diameter is always considered to be a material property constant, which means that the components made from the same material will have the same field diameter. According to this assumption, if the field diameter of a crankshaft is determined, the fatigue load of another type of crankshaft made from the same material can also be determined. This process is detailed hereafter (in Fig 1)

Where $M_e$ is the prediction of the fatigue limit load of the second crankshaft, and $\sigma_b$ and $\sigma_{FI}(A)$ are the fatigue strength of the material and the stress field intensity of the second crankshaft under a given bending moment $M_A$ respectively.

## 3 Results

### 3.1 Case one

According to the analysis in the previous chapter, the first step of the stress field intensity approach is determining the field diameter. Table 1 shows the test data of a given crankshaft, which was assigned the serial number N0. According to the SAFL (Statistical Analysis for Fatigue Limit) theory, the distribution of the fatigue load (at a specified fatigue life) can be expressed by a normal distribution function [32]. Therefore, the fatigue limit load in this case is 5045 N·m.

As shown in Fig 2, in this finite element model, the bending moment is applied on the left side and the boundary conditions are conducted by fastening all the degrees of freedom on the right side. In this case, the material of both crankshafts is a type of high strength alloy steel. The detailed information of this model is in Table 2:

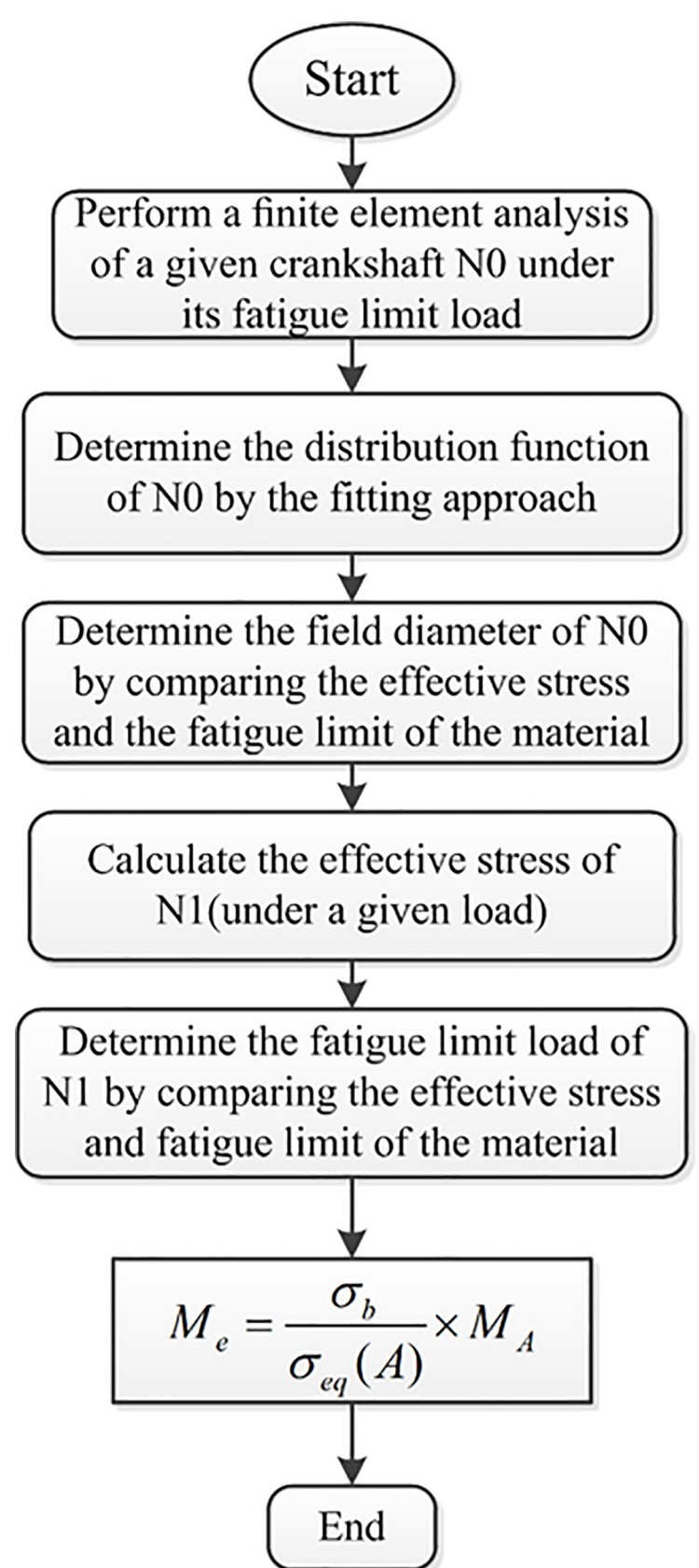

**Fig 1. The process of the prediction.**

**Table 1. Fatigue experiment results of crankshaft N0.**

| Load moment (N·m) | Failure life (cycles) |
|---|---|
| 5352 | 2201350 |
| 5988 | 868299 |
| 6074 | 543448 |
| 5207 | 5464627 |
| 6017 | 779762 |
| 5988 | 1043235 |
| 6278 | 575953 |
| 6133 | 327416 |
| 6104 | 402108 |

In this paper, the FE software Abaqus was used to determine the stress distribution of this crankshaft, and the results are shown in Fig 3.

As shown in Fig 3, the maximum stress point is located at the fillet of the crank pin. Table 3 shows the results of the mesh convergence test, wherein the mesh size is varied from 0.2 mm

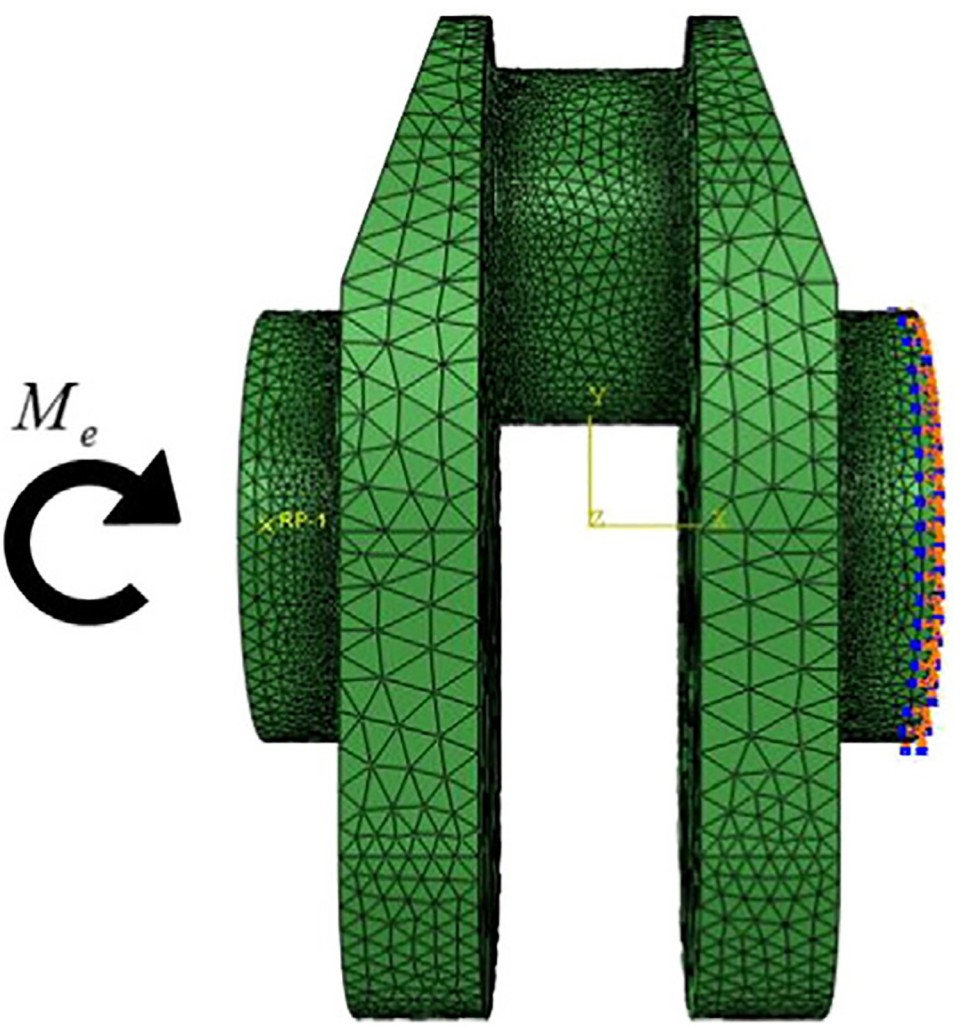

**Fig 2. FE model of the crankshaft.**

**Table 2. Model parameters of crankshaft N0.**

| Model parameter | Value | Model parameter | Value |
|---|---|---|---|
| Young modulus | 210000MPa | Temperature | 20˚C |
| Poisson's ratio | 0.3 | Yield strength | 850MPa |
| Element type | C3D10 | Element number | 240914 |

to 1.5 mm. In this table, clear mesh convergence can be observed, so this element size can be adopted to balance the computation efficiency and the accuracy.

As shown in Table 3, the maximum stress at the fillet is 579 MPa. Then the stress fitting approach mentioned in previous chapter is adopted to fit the stress distribution and the results are in Fig 4 and Table 4:

As shown in Fig 4 and Table 4, the distribution function and the original stress data are in good agreement (the relative difference at each point is less than 1%), so this distribution function can replace the actual stress distribution for further study. The distribution function is expressed as follows:

$$\sigma(r) = \frac{48.96r + 837}{r + 1.442} \tag{3.1}$$

In this paper, the surface treatment technique of the crankshaft is nitriding. The fatigue software Femfat is used to obtain the S-N curve of this material, the result is shown Fig 5:

As shown in Fig 5, the fatigue limit of the material in this case is 523MPa. Therefore, when the value of its stress field intensity is 523 MPa, the corresponding integral range is the field diameter. The express of the function is as follows:

$$\sigma_{FI1}(L = R) = \frac{1}{R} \int_0^R \sigma_1(r)\varphi_1(r)dr = 523MPa \tag{3.2}$$

In this paper, MATLAB software is used to solve the function and the field diameter can be determined to be 0.305mm. Thereafter, a new type of crankshaft (with the serial number N1) is selected as the object of prediction on which a bending moment (1000 N·m) is applied. After

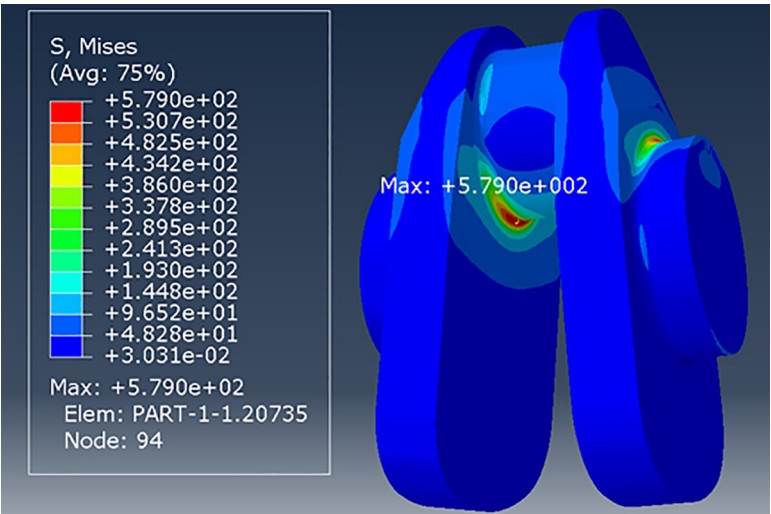

**Fig 3. The stress nephogram of crankshaft N0.**

**Table 3. Mesh convergence test results.**

| Grid size (mm) | Von Mises stress (MPa) | Tresca stress (MPa) | Maximum principal stress (MPa) |
|---|---|---|---|
| 1.5 | 570.4 | 637.6 | 638.1 |
| 1 | 572.9 | 639.1 | 640.6 |
| 0.5 | 578.6 | 639.1 | 646.8 |
| 0.2 | 579.1 | 641.7 | 647.8 |

repeating the fitting approach above to fit the stress distribution in this crankshaft, the corresponding results are shown in Fig 6 and Table 5.

$$\sigma(r) = \frac{-4.016r + 186.6}{r + 0.8844} \tag{3.3}$$

As shown in Fig 6 and Table 5, the relative difference at each node is less than 1%, so this distribution function can accurately reflect the actual stress distribution in crankshaft N1

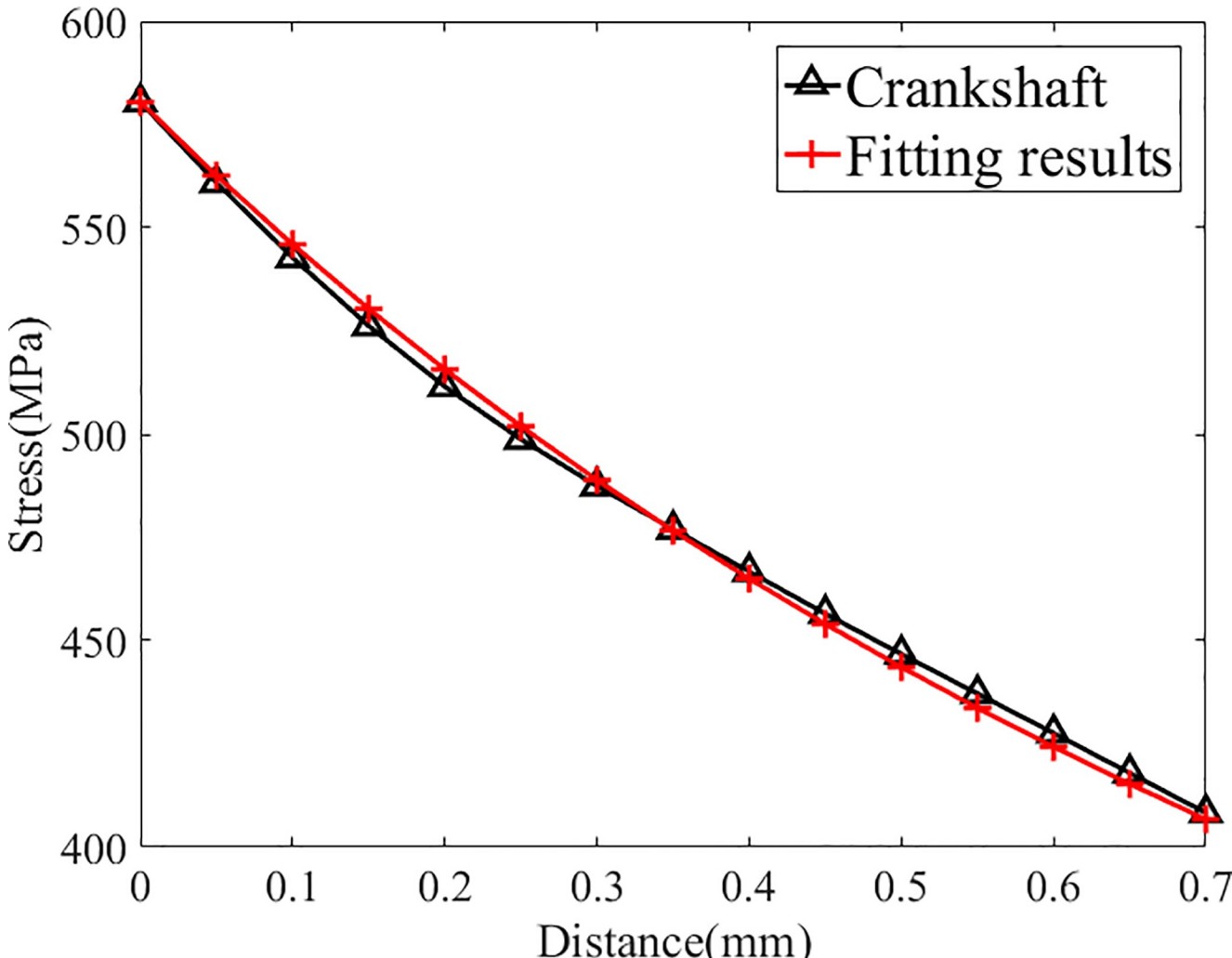

**Fig 4. Fitting results of crankshaft N0(under its fatigue limit load).**

**Table 4. Fitting results of crankshaft N0(under its fatigue limit load).**

| Node number | $\sigma_{FE}$ (MPa) | $\sigma_i$ (MPa) | Relative difference (%) |
|---|---|---|---|
| 1 | 546.27 | 549.23 | 0.54 |
| 2 | 517.19 | 521.47 | 0.83 |
| 3 | 494.13 | 496.63 | 0.51 |
| 4 | 474.93 | 474.27 | 0.14 |
| 5 | 456.55 | 454.04 | 0.55 |
| 6 | 439.09 | 435.64 | 0.79 |
| 7 | 421.83 | 418.85 | 0.71 |
| 8 | 404.47 | 403.45 | 0.25 |

accurately. The distribution function and stress field intensity and the prediction of the fatigue limit load are as follows:

$$\sigma_{FI2}(r = 0.305) = \frac{1}{0.305} \int_0^{0.305} \sigma_2(r)\varphi_2(r)dr = 178.4 MPa \tag{3.4}$$

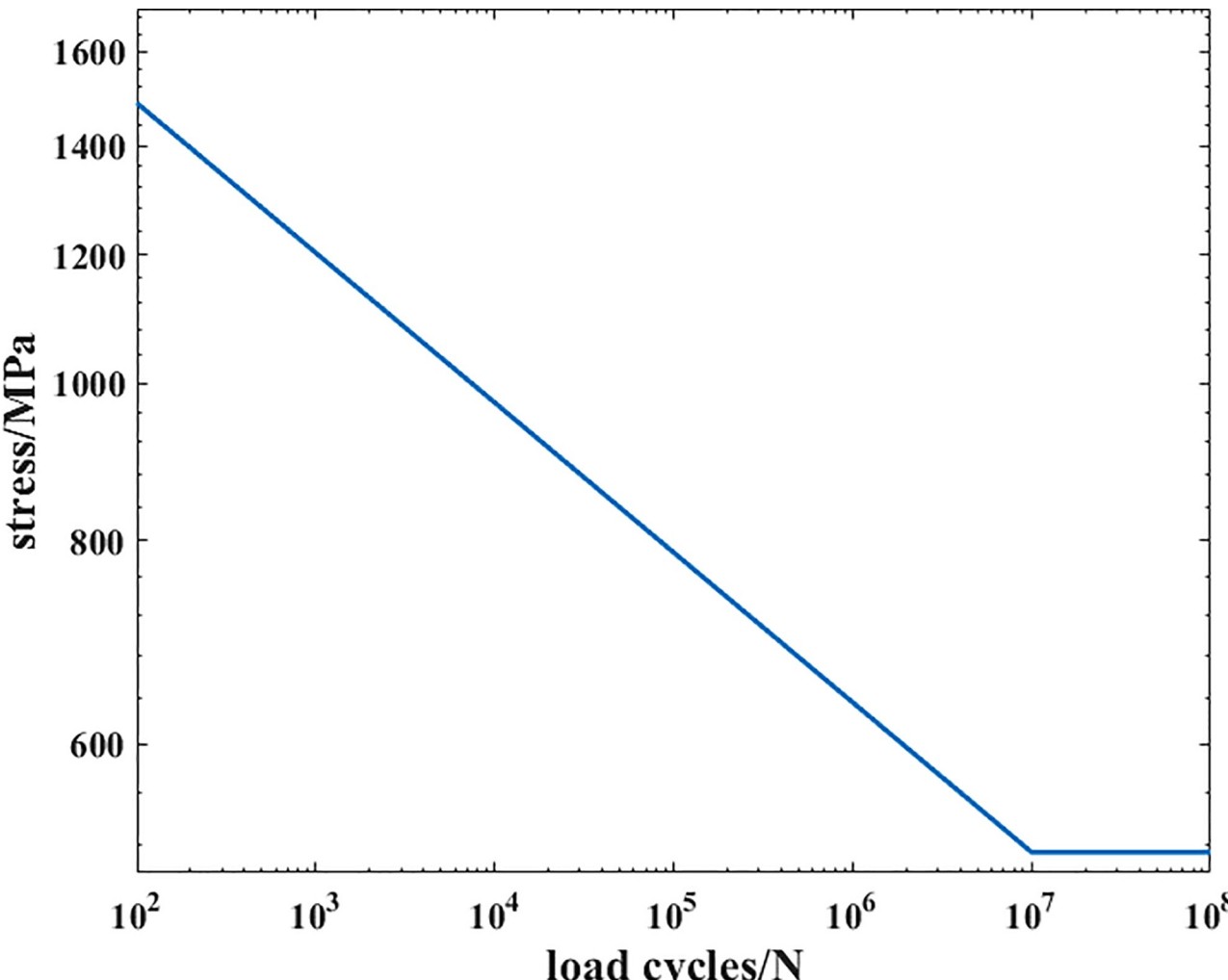

**Fig 5. The S-N curve of the material in the first case.**

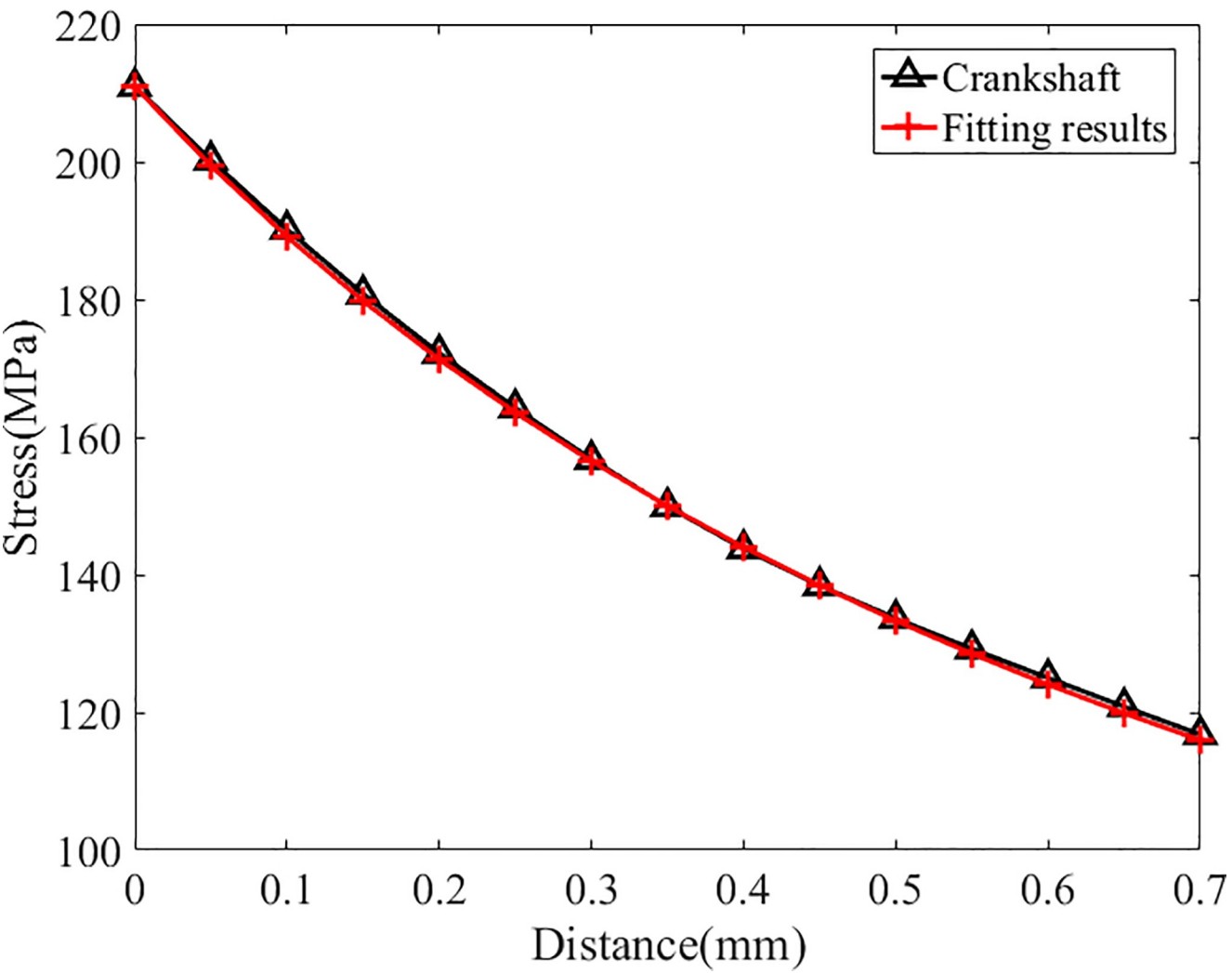

**Fig 6. Fitting results of crankshaft N1(under 1000 N·m bending moment).**

$$M_e = \frac{\sigma_b}{\sigma_{FI}(A)} \times 1000N \cdot m = 2932N \cdot m \tag{3.5}$$

**Table 5. Fitting results of crankshaft N1(under 1000 N·m bending moment).**

| Node number | $\sigma_{FE}$ (MPa) | $\sigma_i$ (MPa) | Relative difference (%) |
|---|---|---|---|
| 1 | 192.14 | 191.09 | 0.54 |
| 2 | 175.53 | 174.6 | 0.53 |
| 3 | 161.18 | 160.67 | 0.32 |
| 4 | 148.63 | 148.76 | 0.09 |
| 5 | 138.43 | 138.46 | 0.02 |
| 6 | 130.1 | 129.45 | 0.49 |
| 7 | 122.49 | 121.52 | 0.79 |
| 8 | 115.24 | 114.48 | 0.66 |

**Table 6. Model parameters of crankshaft C0.**

| Model parameter | Value | Model parameter | Value |
|---|---|---|---|
| Young modulus | 205000MPa | Temperature | 20˚C |
| Poisson's ratio | 0.3 | Yield strength | 880MPa |
| Element type | C3D10 | Element number | 201777 |

## 3.2 Case two

To further illustrate the applicability of our research, another case is selected to repeat the process. In this case, the material of both crankshafts is another type of high strength steel. Detailed information of the model is in Table 6:

The technique described above is repeated to determine the field diameter, and in this case the first crankshaft is assigned the serial number C0. The bending fatigue test results for crankshaft C0 are shown in Table 7:

According to the SAFL method described in last chapter, the fatigue limit load of this crankshaft is 5060 N·m. The approach described above is repeated to obtain the stress distribution under this load, the distribution function is expressed as follows::

$$\sigma(r) = \frac{-17.05r + 785}{r + 0.9043} \tag{3.6}$$

As shown in Fig 7 and Table 8, the distribution function and the original stress data are in good agreement (the relative difference at each point is less than 1.5%), so this distribution function can replace the actual stress distribution for further study. Repeating the approach above to determine the S-N curve in this case, the result is in Fig 8:

Therefore, the field diameter of this crankshaft can be determined as follows:

$$\sigma_{FI1}(L = R) = \frac{1}{R} \int_0^R \sigma_1(r)\varphi_1(r)dr = 640MPa \tag{3.7}$$

Using the MATLAB, the field diameter in this case can be determined to be 0.52mm. Therefore, another type of crankshaft (serial number C1) is selected as the object of study, on which a 1000 $N·m$ bending moment is applied. After repeating the fitting approach above, the function can be determined and the corresponding results are shown in Fig 9:

$$\sigma(r) = \frac{12.17r + 171.2}{r + 1.292} \tag{3.8}$$

**Table 7. Fatigue experiment results of crankshaft C0.**

| Bending moment /$N·m$ | Fatigue life |
|---|---|
| 4300 | 1672344 |
| 4700 | 244397 |
| 4600 | 279267 |
| 4500 | 462782 |
| 4200 | 6812699 |
| 4400 | 2274229 |
| 4150 | 5047681 |
| 4400 | 1137654 |

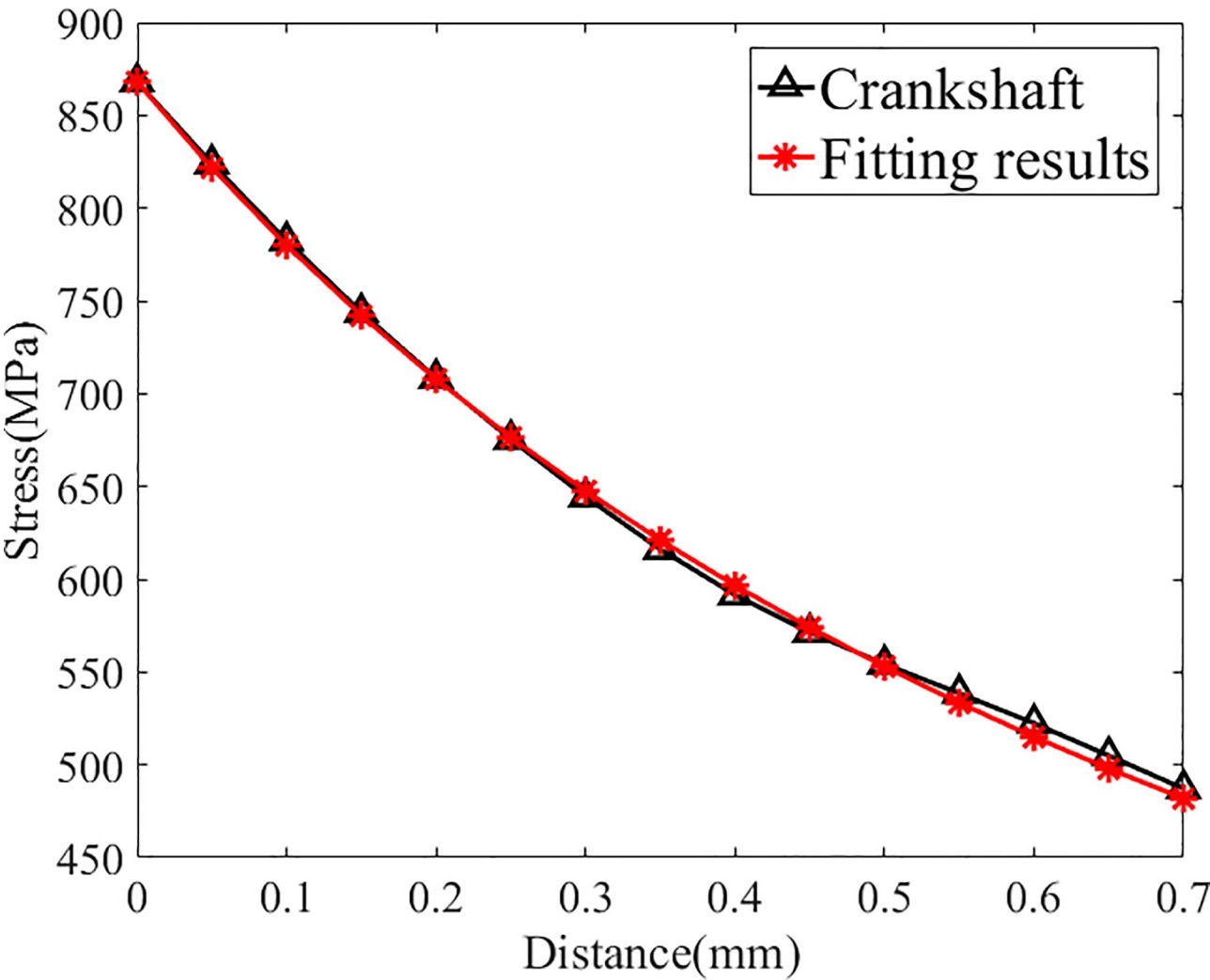

**Fig 7. Fitting results of crankshaft C0(under its fatigue limit load).**

As shown in Fig 9 and Table 9, the relative difference at each node is less than 1%, so this distribution function can accurately reflect the actual stress distribution in crankshaft C1 accurately. The distribution function and stress field intensity and the prediction of the fatigue

**Table 8. Fitting results of crankshaft C0(under its fatigue limit load).**

| Node number | $\sigma_{FE}$ (MPa) | $\sigma_i$ (MPa) | Relative difference (%) |
|:---:|:---:|:---:|:---:|
| 1 | 790.49 | 787.94 | 0.32 |
| 2 | 722.11 | 721.13 | 0.14 |
| 3 | 662.76 | 664.55 | 0.27 |
| 4 | 610.89 | 616.03 | 0.84 |
| 5 | 571.39 | 573.96 | 0.45 |
| 6 | 541.65 | 537.13 | 0.83 |
| 7 | 512.02 | 504.62 | 1.44 |
| 8 | 480.01 | 475.72 | 0.89 |

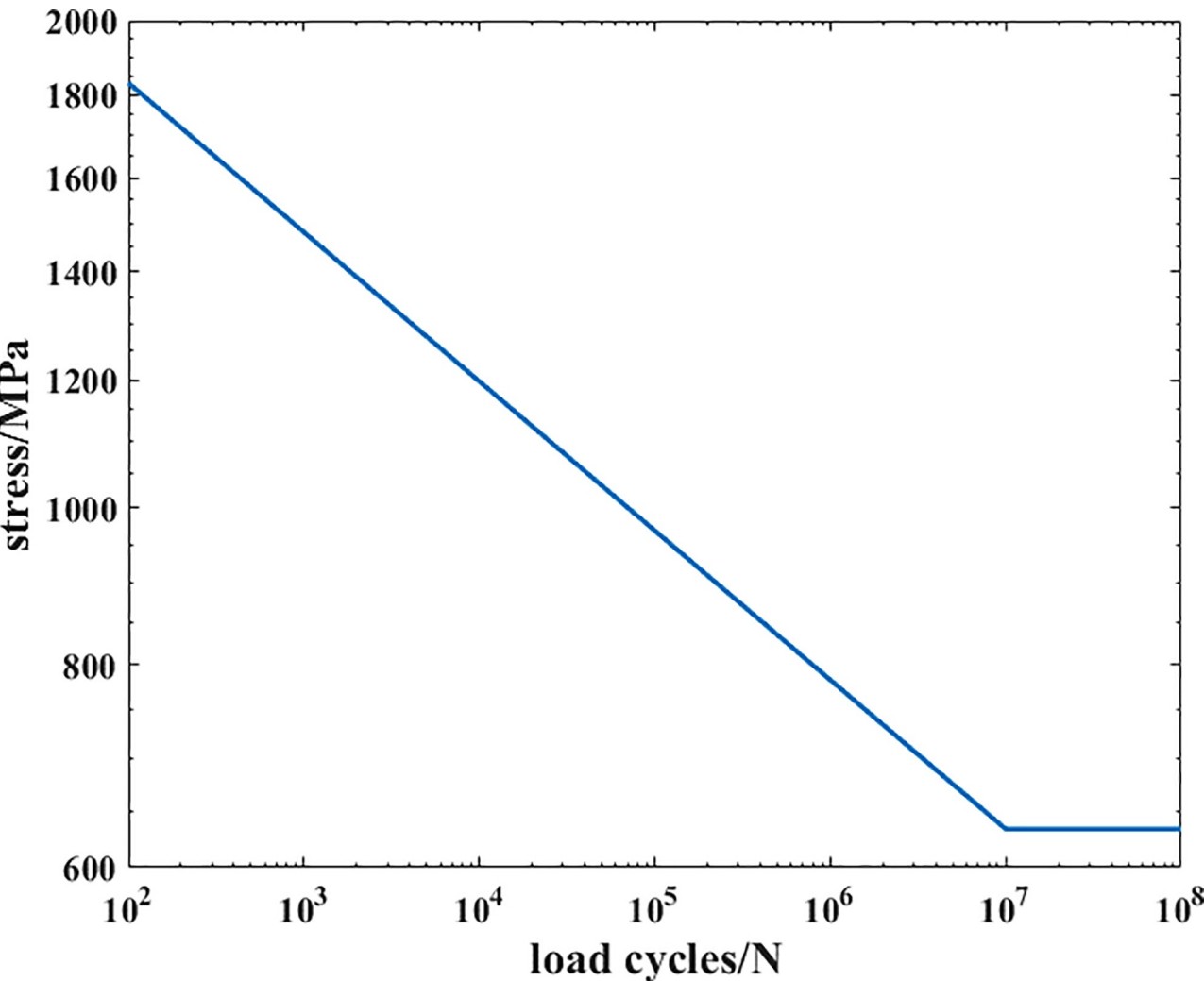

**Fig 8. The S-N curve of the material in the second case.**

limit load are as follows:

$$\sigma_{FI2}(r = 0.52) = \frac{1}{0.52} \int_0^{0.52} \sigma_2(r)\varphi_2(r)dr = 109.2 MPa \tag{3.9}$$

$$M_e = \frac{\sigma_b}{\sigma_{FI}(A)} \times 1000 N \cdot m = 5860 N \cdot m \tag{3.10}$$

## 4 Experimental verification

To check the accuracy of the prediction, corresponding experimental verification is necessary. Fig 10 shows the bending fatigue test equipment that consists of the electromagnetic vibration exciter, the master arm, the slave arm, the acceleration transducer, and the foundation bed. During the experiment process, the crankshaft and the connected arms are vertically supported by springs, and the excitation force is generated by rotating the eccentric with the motor. In this way, a cyclic bending moment is applied on the crankshaft for fatigue testing. During the experiment process, the crack is expected to appear at the fillet of the crankshaft

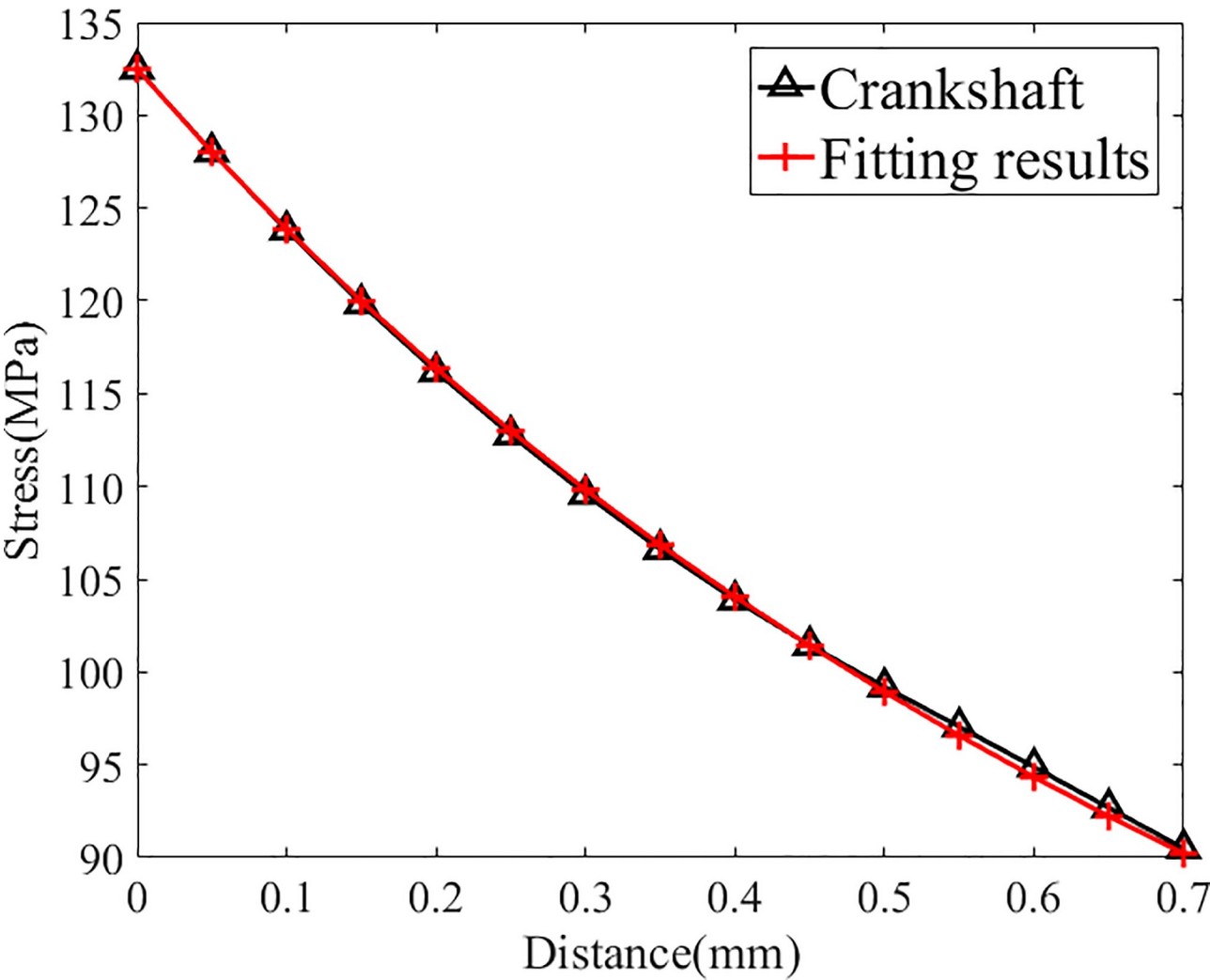

**Fig 9. Stress distribution of crankshaft C1(under 1000 N·m).**

and the stiffness of the system decreases. As a result of this, the responsive acceleration and the amplitude of the load may increase if the frequency of the exciter remains unchanged. To avoid this unwanted situation, the rotation speed of the exciter will decrease accordingly. When the speed decreases to a certain level, the crankshaft is considered broken. The serial

**Table 9. Stress distribution of crankshaft C1 (under 1000 N·m).**

| Node number | $\sigma_{FE}$ (MPa) | $\sigma_i$ (MPa) | Relative difference (%) |
|:---:|:---:|:---:|:---:|
| 1 | 124.64 | 124.66 | 0.02 |
| 2 | 117.62 | 117.76 | 0.14 |
| 3 | 111.48 | 111.71 | 0.19 |
| 4 | 106.05 | 106.28 | 0.21 |
| 5 | 101.44 | 101.42 | 0.02 |
| 6 | 97.48 | 97.03 | 0.46 |
| 7 | 93.56 | 93.06 | 0.55 |
| 8 | 89.63 | 89.44 | 0.21 |

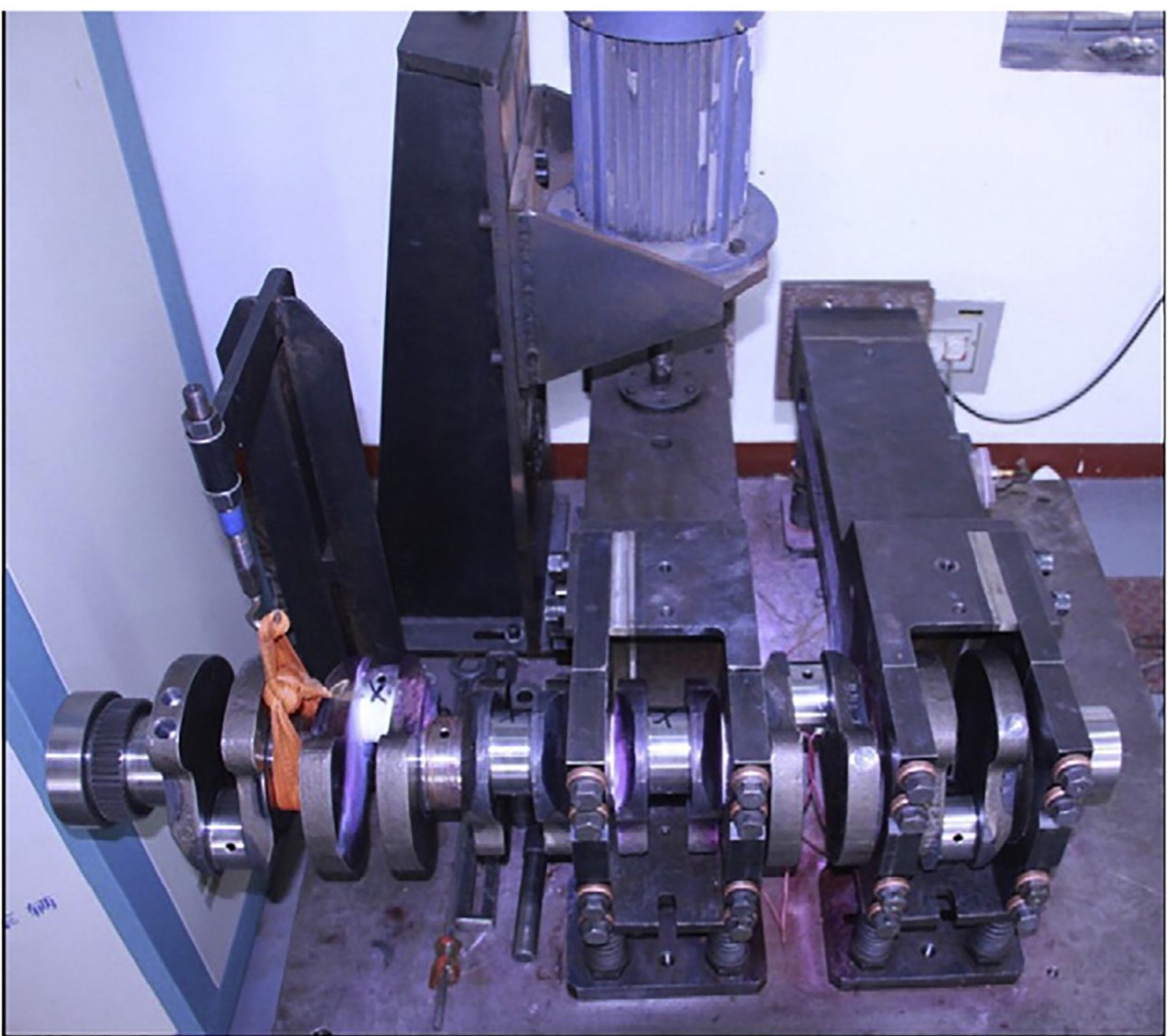

**Fig 10. Crankshaft bending fatigue test system.**

**Table 10. Fatigue experiment results of crankshaft N1.**

| Bending moment /$N \cdot m$ | Fatigue life |
|---|---|
| 3335 | 6711179 |
| 3436 | 6187261 |
| 3881 | 502460 |
| 3901 | 548588 |
| 3416 | 6142771 |
| 3557 | 3509554 |
| 3941 | 273493 |
| 3949 | 613944 |

**Table 11. Fatigue experiment results of crankshaft C1.**

| Bending moment /$N·m$ | Fatigue life |
|---|---|
| 5988 | 868299 |
| 6074 | 543448 |
| 5207 | 5464627 |
| 6017 | 779762 |
| 5988 | 1043235 |
| 6278 | 575953 |
| 6133 | 327416 |
| 6104 | 402108 |
| 5497 | 3318128 |

number of the experiment standard used in this paper is QC-T637-2000. The corresponding results are shown in Tables 10 and 11.

Using the SAFL approach, the fatigue limit loads of t crankshafts N1 and C1 are 3335 $N·m$ and 5125 $N·m$ respectively. Table 12 shows the errors of the two cases and a clear conclusion can be determined that the conventional stress field intensity approach may result in some errors (over 10%) in this occasion.

According to the literature [33], among the main structural parameters of the crankshaft, the fillet radius of the main journal obviously affects the fatigue. Table 13 shows the main structural parameters of all the crankshafts in both cases, which reveals that the error may be attributed to the structural differences between the datum crankshaft and the object of prediction.

## 5 Discussion and model modification

### 5.1 The modified model

In the previous chapter, we evaluated the applicability of the conventional stress field intensity approach in the fatigue analysis of several sets of crankshafts, wherein a primary conclusion indicated that the field diameter of a crankshaft should not be considered as a constant. However, until now, no standard modified models has been proposed.

According to the definition of the stress field intensity approach, the fatigue life of a given component depends on the stress field around the hot point (usually the maximum stress point). Among the common parameters, the stress gradient is usually used to evaluate the distribution property. Therefore, this paper proposes a relative stress gradient based modification, the corresponding model is as follows:

$$R = C · M \tag{5.1}$$

where $R$ is the diameter of the field, $M$ is a constant that is dependent on the material properties, and $C$ is the value of the maximum relative stress gradient. Based on the fitting results, the relative stress gradient can be determined as follows:

$$C = \frac{S_{Gmax}}{\sigma_{max}} = \frac{\frac{d\sigma}{dr}\big|_{r=0}}{\sigma(r=0)} = \frac{AC - B}{BC} \tag{5.2}$$

**Table 12. Prediction errors based on the conventional stress field intensity approach.**

| N1 | C1 |
|---|---|
| 12.1% | 14.3% |

**Table 13. The main structural parameters of all the crankshafts.**

| Case one | | | Case two | | |
|---|---|---|---|---|---|
| **Serial number** | **N0** | **N1** | **Serial number** | **C0** | **C1** |
| Spindle diameter | 100mm | 82mm | Spindle diameter | 82mm | 100mm |
| Crankpin diameter | 82mm | 68mm | Crankpin diameter | 68mm | 83mm |
| Fillet radius | 29mm | 26mm | Fillet radius | 26mm | 28mm |
| Overlap | 5mm | 3mm | Overlap | 3mm | 5mm |
| Crank width | 26mm | 6mm | Crank width | 6mm | 16mm |

Based on this equation, the maximum relative stress gradient of all the crankshafts can be determined, the results are in Table 14:

### 5.2 Case one

According to the modified model proposed in this paper, the field diameter of crankshaft N1 will be modified as follows:

$$R_2 = C_2 \cdot M = \frac{C_2 R_1}{C_1} = 0.56mm \tag{5.3}$$

Based on the modified diameter and the distribution function, the fatigue limit load of crankshaft N1 can be determined as follows:

$$\sigma_{FI2}(r = 0.555) = \frac{1}{0.56} \int_0^{0.56} \sigma_2(r)\varphi_2(r)dr = 151.6MPa \tag{5.4}$$

$$M_e = \frac{\sigma_b}{\sigma_{FI}(A)} \times 1000N \cdot m = 3450N \cdot m \tag{5.5}$$

### 5.3 Case two

According to the modified model proposed in this paper, the field diameter of crankshaft N1 will be modified as follows:

$$R_2 = C_2 \cdot M = \frac{C_2 R_1}{C_1} = 0.31mm \tag{5.6}$$

**Table 14. The relative stress gradient calculation of all the crankshafts.**

| Serial number | The relative stress gradient |
|---|---|
| N0 | 0.63mm$^{-1}$ |
| N1 | 1.15mm$^{-1}$ |
| CO | 1.12mm$^{-1}$ |
| C1 | 0.64mm$^{-1}$ |

**Table 15. Prediction errors based on the modified stress field intensity approach.**

| N1 | C1 |
|---|---|
| 3.2% | 5.8% |

Based on the modified diameter and the distribution function, the fatigue limit load of crankshaft C1 can be determined as follows:

$$\sigma_{FI2}(r = 0.3) = \frac{1}{0.3} \int_0^{0.3} \sigma_2(r)\varphi_2(r)dr = 118.2 MPa \tag{5.7}$$

$$M_e = \frac{\sigma_b}{\sigma_{FI}(A)} \times 1000N \cdot m = 5423N \cdot m \tag{5.8}$$

## 5.4 Comparison and discussion

From the two equations above, we can find that for both cases, the predictions based on the modified stress field intensity approach are quite different from those based on the conventional stress field intensity approach. Comparing them with the experimental results above, corresponding errors are in Table 15:

Table 15 clearly shows that compared with the conventional stress field intensity model, the modified model provides much more accurate predictions. Moreover, the error rate in the modified model can already meet the usual engineering demands(less than 10%), so this approach seems to be more suitable for actual engineering applications.

## 6 Conclusion

(1) A previous study revealed that, the conventional stress field intensity approach cannot be used to investigate the fatigue properties of parts with complicated shaped due to the inability of easily determining the stress distribution function easily. This paper proposed a combined finite element analysis and inverse function approach in to fit the crankshaft stress distribution, which can replace the actual stress state during fatigue research.

(2) A modified stress field intensity approach, which is based on the product of the field diameter and the relative stress gradient, was proposed and applied for crankshaft fatigue limit load prediction. The results showed that compared with the conventional model, this new approach exhibited better predictive accuracy.

## Supporting information

**S1 File. Nomenclature.**
(DOCX)

## Author Contributions

**Conceptualization:** SongSong Sun.

**Data curation:** SongSong Sun.

**Formal analysis:** SongSong Sun.

**Funding acquisition:** SongSong Sun.

**Investigation:** SongSong Sun.

**Methodology:** SongSong Sun.

**Project administration:** SongSong Sun.

**Resources:** SongSong Sun.

**Software:** SongSong Sun.

**Supervision:** SongSong Sun.

**Validation:** SongSong Sun.

**Visualization:** SongSong Sun.

**Writing – original draft:** SongSong Sun.

**Writing – review & editing:** SongSong Sun.

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
