## [Decision Letter · Decision Letter 0]

21 Apr 2020

PONE-D-20-07563

A new stress field intensity model and its application in component high cycle fatigue research

PLOS ONE

Dear Dr. Sun,

Thank you for submitting your manuscript to PLOS ONE. After careful consideration, we feel that it has merit but does not fully meet PLOS ONE’s publication criteria as it currently stands. Therefore, we invite you to submit a revised version of the manuscript that addresses the points raised during the review process.

As summarized in the review reports, this paper requires major revisions and extensive modifications. Please, address all the comments made by the reviewers, in particular, put this paper into the correct context of the current fatigue theories as noticed by reviewers 1, 3 and 4.

We would appreciate receiving your revised manuscript by Jun 05 2020 11:59PM. To enhance the reproducibility of your results, we recommend that if applicable you deposit your laboratory protocols in protocols.io, where a protocol can be assigned its own identifier (DOI) such that it can be cited independently in the future. For instructions see: http://journals.plos.org/plosone/s/submission-guidelines#loc-laboratory-protocols

We look forward to receiving your revised manuscript.

Kind regards,

Antonio Riveiro Rodríguez, PhD

Academic Editor

PLOS ONE

Reviewers' comments:

Reviewer's Responses to Questions

**Comments to the Author**

1. Is the manuscript technically sound, and do the data support the conclusions?

Reviewer #1: No

Reviewer #2: Yes

Reviewer #3: No

Reviewer #4: Yes

2. Has the statistical analysis been performed appropriately and rigorously? 

Reviewer #1: No

Reviewer #2: N/A

Reviewer #3: No

Reviewer #4: N/A

3. Have the authors made all data underlying the findings in their manuscript fully available?

Reviewer #1: Yes

Reviewer #2: Yes

Reviewer #3: No

Reviewer #4: Yes

4. Is the manuscript presented in an intelligible fashion and written in standard English?

Reviewer #1: No

Reviewer #2: Yes

Reviewer #3: Yes

Reviewer #4: Yes

5. Review Comments to the Author

Reviewer #1: The paper does not use appropriate scientific description of the problem and appropriate terminology. For example, what does “field diameter” mean? In the introduction, the first sentence is badly written and it refers to impacts occurring in combustion engines. I would’t call the loads applied on the crankshaft as “impacts” because it is not the case, otherwise you should employ a completely different structural integrity approach.

Apart from that, the paper simply ignores all the relevant literature concerning the fatigue criteria generally used for these types of problems. On top of that, it is ignored completely the multiaxiality issue of this case-study.

Due to these major shortcomings, I cannot suggest acceptance of this paper.

Reviewer #2: In this manuscript, the field diameter of a given crankshaft was determined based on its limit stress state and a stress distribution fitting approach. Then, this parameter was used to predict the high-cycle bending fatigue limit load of a new crankshaft composed of the same material. Furthermore, the proposed model in this manuscript provides higher accuracy in fatigue load prediction. The topic is interesting.

• This topic was previously studied by the author, Ref. # 1. Therefore, the author should explain the novelty in this manuscript.

• Some previous important contributions in the literature are not included in both introduction and discussion, See for example the following Refs.:

a. "Significance of crack tip plasticity to early notch fatigue crack growth", Int. J. of Fatigue, Vol. 26, No. 2, pp. 173-182, 2004.

b. "Mode I notch fatigue crack growth behaviour under constant amplitude loading and due to the application of a single tensile overload", Int. J. of Fatigue, Vol. 26, No. 2, pp. 183-192, 2004.

c. "Deformation behaviour at the tip of physically short fatigue crack due to a single overload", Fatigue Fract. Engng Mater. Struct., Vol. 22,145-151, 1999.

These references have done deep investigation in the related topic, they consists of key queries which are required for response by the authors.

• The English language should be polished throughout the manuscript, for example:

o In Abstract: this statement should be revised " The new model proposed in this paper provided higher accuracy in quick fatigue load prediction,"

o In Introduction,

third paragraph: " In recent years," Refs. # 6 & 7 were published in 1993 & 1995 !!!!!!!

Howevwe, the !!!!

theory of critical distance(TCD) approach. This approach should be refer to the scholar who was suggested it (Taylor D. The theory of critical distances: a new perspective in fracture mechanics. Oxford, UK: Elsevier; 2007)

o In Method: Nomenclature should be moved in a appropriate location (After Abstract or After References)

• Abstract should be extended.

Reviewer #3: This paper works on the application of the stress field intensity theory for quick component fatigue limit load predictions. There are many issues for the present study since many details about its experimental and simulation model are missing.

The equation of stress field intensity is provided. However, the authors make an assumption of the method and uses experimental results to fit the assumption. Nevertheless, there are no material parameters, no temperature, no boundary condition, and no model dimensions that are defined or provided. In addition, the paper does not provide FEM detail such as element types, which will have a huge influence on the accuracy of the simulation results.

Additional detailed comments:

1. In introduction section, the author introduces the stress field intensity method and compares it with the TCD method. The author should address more details about this stress field intensity method. For example, how does it work, why does this paper choose this method other than TCD which is similar to the stress field intensity method?

2. In section 2.2. The Eq. (2.5) is used as an assumed stress distributed function. The author should provide details about this assumption. If this assumption comes from other studies, the reference should be provided. If this equation is assumed by the author, more details and the limitation of this assumption should be provided.

3. In Section 3.1, the author gives the test results (Tab. 1) and the corresponding finite element model (Fig. 2). The model details should be provided, such as material properties, temperature, boundary condition, and model dimensions. Without this data, such experiments and simulations cannot be repeated by others.

4. What is the difference between Case one and Case two?

Reviewer #4: In this paper, a new stress field intensity model and its application in component high cycle fatigue research is proposed. The overall balance and structure of the paper is good; however, before publication the following improvements are recommended:

- The Introduction needs to be significantly improved. There are certain research groups (Marsavina et al., Linul et al., Berto et al., Sadowski et al., etc.) dealing with the fatigue property prediction and not only. Please refer to their works in the Introduction.

- The standard deviations should be added in the Tables.

- The error bars should be added in the Figures 4-7.

- It is recommended to draw Wöhler curves.

- The used standard must be presented.

- References must be uniformized. Some journals are abbreviated and others are not

- The literature review should consider more in details the recent developments published in the PLOS ONE journal, showing a continuity between the present paper and those reported in the literature on similar topics.

- English is not the native language of this Reviewer; however, there are a number of spelling errors and other mistakes in grammar. Therefore, proof-reading of the paper for correct English would be highly advisable.

6. PLOS authors have the option to publish the peer review history of their article (what does this mean?). If published, this will include your full peer review and any attached files.

Reviewer #1: No

Reviewer #2: Yes: Hossam El-Din M. Sallam

Reviewer #3: No

Reviewer #4: No

---

## [Author Response · Author response to Decision Letter 0]

6 May 2020

Replies to the reviewers’ comments:

Reviewer #1:

1. For example, what does “field diameter” mean?

Response：the definition of field diameter has been introduced in the introduction part (under equation 1.4)

2. I would’t call the loads applied on the crankshaft as “impacts” because it is not the case, otherwise you should employ a completely different structural integrity approach.

Response: the load has been renamed as dynamic load

3. On top of that, it is ignored completely the multiaxiality issue of this case-study.

Response: the stress field intensity approach is just considered to be an effective tool in researching the multiaxial fatigue problems, we have explained in the revised manuscript.

Reviewer #2:

1. This topic was previously studied by the author, Ref. # 1. Therefore, the author should explain the novelty in this manuscript.

Response: the novelty in this manuscript is the new modified stress field model, we have explained in the manuscript.

2. Some previous important contributions in the literature are not included in both introduction and discussion, See for example the following Refs.:

a. "Significance of crack tip plasticity to early notch fatigue crack growth", Int. J. of Fatigue, Vol. 26, No. 2, pp. 173-182, 2004.

b. "Mode I notch fatigue crack growth behaviour under constant amplitude loading and due to the application of a single tensile overload", Int. J. of Fatigue, Vol. 26, No. 2, pp. 183-192, 2004.

c. "Deformation behaviour at the tip of physically short fatigue crack due to a single overload", Fatigue Fract. Engng Mater. Struct., Vol. 22,145-151, 1999.

Response: these articles have been added to the reference.

3. The English language should be polished throughout the manuscript

Response: the whole manuscript has been edited by the AJE company.

4. Abstract should be extended

Response: the abstract has been extended

Reviewer #3:

1. In introduction section, the author introduces the stress field intensity method and compares it with the TCD method. The author should address more details about this stress field intensity method. For example, how does it work, why does this paper choose this method other than TCD which is similar to the stress field intensity method?

Response: the introduction of the stress field intensity approach has been removed to the introduction section, and the reason of the choice has been explained (the feasibility in researching the multiaxial fatigue problems)

2. In section 2.2. The Eq. (2.5) is used as an assumed stress distributed function. The author should provide details about this assumption. If this assumption comes from other studies, the reference should be provided. If this equation is assumed by the author, more details and the limitation of this assumption should be provided.

Response: the distribution function comes from Adrea’s article, which has been added in the manuscript.

3. In Section 3.1, the author gives the test results (Tab. 1) and the corresponding finite element model (Fig. 2). The model details should be provided, such as material properties, temperature, boundary condition, and model dimensions. Without this data, such experiments and simulations cannot be repeated by others.

Response: the detailed information of the finite element model has been added in the manuscript, combined with the structural parameters in Tab.13, the simulation can be repeated.

4. What is the difference between Case one and Case two?

Response: the difference is the material, which has been stated by the S-N curves in Fig.5 and Fig.8.

Reviewer #4:

1. The Introduction needs to be significantly improved. There are certain research groups (Marsavina et al., Linul et al., Berto et al., Sadowski et al., etc.) dealing with the fatigue property prediction and not only. Please refer to their works in the Introduction.

Response: these works have been added in the introduction part.

2. The standard deviations should be added in the Tables.

Response: we are sorry but we can’t understand which tables should be added with the standard deviations, because the parameters in the tables are just finite element analysis results, structural parameters, and experiment results. They are not statistical data. Please indicate this more clearly

3. The error bars should be added in the Figures 4-7.

Response: the errors of the fitting results are too small to be expressed by curves, so we added four more tables(4,5,8,9) to explain them more clearly

4. It is recommended to draw Wöhler curves.

Response: the curves have been added in both cases.

5. The used standard must be presented.

Response: we have added the experiment standard in the manuscript.

6. References must be uniformized. Some journals are abbreviated and others are not.

Response: the references have been uniformized.

7. The literature review should consider more in details the recent developments published in the PLOS ONE journal, showing a continuity between the present paper and those reported in the literature on similar topics.

Response: some articles have been added(ref.8&12)

8. English is not the native language of this Reviewer; however, there are a number of spelling errors and other mistakes in grammar. Therefore, proof-reading of the paper for correct English would be highly advisable.

Response: the whole manuscript has been edited by the AJE company.

---

## [Decision Letter · Decision Letter 1]

15 Jun 2020

A new stress field intensity model and its application in component high cycle fatigue research

PONE-D-20-07563R1

Dear Dr. Sun,

We are pleased to inform you that your manuscript has been judged scientifically suitable for publication and will be formally accepted for publication once it complies with all outstanding technical requirements.

With kind regards,

Antonio Riveiro Rodríguez, PhD

Academic Editor

PLOS ONE

Reviewers' comments:

Reviewer's Responses to Questions

**Comments to the Author**

1. If the authors have adequately addressed your comments raised in a previous round of review and you feel that this manuscript is now acceptable for publication, you may indicate that here to bypass the “Comments to the Author” section, enter your conflict of interest statement in the “Confidential to Editor” section, and submit your "Accept" recommendation.

Reviewer #1: (No Response)

Reviewer #2: All comments have been addressed

Reviewer #3: All comments have been addressed

Reviewer #4: All comments have been addressed

2. Is the manuscript technically sound, and do the data support the conclusions?

Reviewer #1: Partly

Reviewer #2: Yes

Reviewer #3: Yes

Reviewer #4: (No Response)

3. Has the statistical analysis been performed appropriately and rigorously? 

Reviewer #1: No

Reviewer #2: N/A

Reviewer #3: Yes

Reviewer #4: (No Response)

4. Have the authors made all data underlying the findings in their manuscript fully available?

Reviewer #1: (No Response)

Reviewer #2: Yes

Reviewer #3: Yes

Reviewer #4: (No Response)

5. Is the manuscript presented in an intelligible fashion and written in standard English?

Reviewer #1: (No Response)

Reviewer #2: Yes

Reviewer #3: Yes

Reviewer #4: (No Response)

6. Review Comments to the Author

Reviewer #1: The authors have only partially and superficially addressed my concerns. Their effort in answering my questions was very little.

Reviewer #2: The authors have successfully addressed all my comments. Therefore, I recommend the publication of this manuscript.

Reviewer #3: The authors have provided more detailed parameters for the FE model. The comments are addressed and could be accepted.

Reviewer #4: (No Response)

7. PLOS authors have the option to publish the peer review history of their article (what does this mean?). If published, this will include your full peer review and any attached files.

Reviewer #1: No

Reviewer #2: Yes: Hossam El-Din M. Sallam

Reviewer #3: No

Reviewer #4: No

---

## [Editor Report · Acceptance letter]

29 Jun 2020

PONE-D-20-07563R1 

A new stress field intensity model and its application in component high cycle fatigue research 

Dear Dr. Sun:

I'm pleased to inform you that your manuscript has been deemed suitable for publication in PLOS ONE. Congratulations! Your manuscript is now with our production department. 

Kind regards, 

on behalf of

Dr. Antonio Riveiro Rodríguez 

Academic Editor

PLOS ONE